# Bearing Fault Diagnosis Based on a Hybrid Classifier Ensemble Approach and the Improved Dempster-Shafer Theory

**DOI:** 10.3390/s19092097

**Published:** 2019-05-06

**Authors:** Yanxue Wang, Fang Liu, Aihua Zhu

**Affiliations:** 1Beijing Key Laboratory of Performance Guarantee on Urban Rail Transit Vehicles, Beijing University of Civil Engineering and Architecture, Beijing 100044, China; liudhuanglf@gmail.com (F.L.); zhuaihua@bucea.edu.cn (A.Z.); 2School of Mechanical and Electrical Engineering, Guilin University of Electronic Technology, Guilin 541004, China

**Keywords:** rolling element bearing, hybrid classifier ensemble, Dempster-Shafer evidence theory, fuzzy preference relations

## Abstract

Bearing fault diagnosis of a rotating machine plays an important role in reliable operation. A novel intelligent fault diagnosis method for roller bearings has been developed based on a proposed hybrid classifier ensemble approach and the improved Dempster-Shafer theory. The improved Dempster-Shafer theory well considered the combination of unreliable evidence sources, the uncertainty information of basic probability assignment, and the relative credibility of the evidence on the weights in the process of decision making under the framework of fuzzy preference relations, which can effectively deal with conflicts of the evidences and then well improve the diagnostic accuracy for the hybrid classifier ensemble. The effectiveness of the improved Dempster-Shafer theory has been verified via a numerical example. In addition, deep neural networks, a support vector machine, and extreme learning machine techniques have been utilized in the single-stage classification based on singular spectrum entropy, power spectrum entropy, time-frequency entropy, and wavelet packet energy spectrum entropy in this work. Performances of the proposed hybrid ensemble classifier has been demonstrated on a bearing test-rig, compared with the original Dempster-Shafer theory. It can be found that the overall error rate can be greatly reduced with the hybrid ensemble classifier and the improved Dempster-Shafer theory.

## 1. Introduction

Rolling element bearings are the key components widely used in rotating machines. A sudden breakdown of the mechanical system or even a severe catastrophe, may be caused due to an unexpected failure of the rolling element bearings. Therefore, many bearing fault diagnosis methods have been developed based on vibration signal analysis and feature extraction [1,2,3]. However, some of them are performed manually with low efficiency by means of knowledge and experiences of experts, which are not practical in real applications. Thus, there is still growing attention towards the development of bearing intelligent fault diagnosis techniques. For example, a novel intelligent fault diagnosis method has been proposed based on the affinity propagation clustering algorithm and the adaptive feature selection technique [4]. Qin et al. [5] proposed a model for fault diagnosis of gearboxes in wind turbines based on deep belief networks (DBNs), using improved logistic sigmoid units and the impulsive signatures. In addition, a three-stage intelligent fault diagnosis clustering technique has been proposed for the industrial process monitoring [6]. Generally, the diagnosis results achieved by using a single-stage classifier may still be precarious [7,8,9,10]. According to Wolpert’s theorem, there is not a single classifier approach that can be successfully applied for all pattern recognition tasks since each has its own domain of competence [11].

Nowadays, many different combinations of several different learning algorithms, such as the hybrid or ensemble systems, have been highlighted as a hot topic and promising trend in the fields of pattern recognition. The hybrid intelligent systems offer many alternatives for unorthodox handling of realistic increasingly complex problems, involving ambiguity, uncertainty, and high-dimensionality of data [12]. Nevertheless, the accuracy of the existing techniques needs to be further improved, since the structure of rotating machinery becomes increasingly complicated. Therefore, a novel hybrid classifier ensemble (HCE) algorithm has been developed in this work, which can perform fault diagnosis under an improved framework of information fusion.

Actually, there are various strategies for information fusion, such as the simple voting procedure [13]. The Dempster-Shafer theory (DST) has been also widely used as a combining decision method due to its uncertainty processing ability [14]. In recent years, DST has attracted lots of attention and has been used in fault diagnosis for different industrial equipment. For example, a fusion approach was proposed for fault diagnosis of roller bearing in the aeroengine based on *n*-dimensional characteristic parameter distance [15]. Since a hybrid technique can substantially increase the accuracy of fault detection, DST combined with Support Vector Machine (SVM) has been applied for bearing multi-fault diagnosis [16]. A fault diagnosis method was proposed for the reactor coolant system of a nuclear power plant based on DST and fuzzy function in reference [17]. DST is well suitable for information fusion, but it may generate counter-intuitive results for highly conflicting and unreliable pieces of evidence [18,19]. Thus, conflict management has always been an unavoidable problem in information fusion using DST, which is also the main limitation of DST. To solve this issue, many improved versions of DST have been proposed, such as the average approach in reference [20], the weighted average based on the evidence distance in reference [21], and the vector space introduced in reference [22]. Most of the available methods employed distance of the evidences as a critical factor to determine the weights, such as the Jousselme distance [23] and the MaxDiff distance [24]. Then, the support degrees of the evidences can be adjusted and be used to generate the appropriate weights with regard to the evidences. It can be found that a bigger weight is set to the reliable evidence and a smaller weight is set to the unreliable evidence. Although these techniques can reduce the influence of the unreliable evidence, they rarely consider the effects of the uncertain information of the evidences.

Many fuzzy modeling approaches have been successfully utilized in various applications in the past decades, since fuzzy sets technique also plays an important role in the decision-making process and can deal well with uncertain information. Qian etc. [25] successfully utilized the advantages of group decision making via fuzzy preference. The fuzzy preference relations (FPR) has been constructed for multiple pieces of evidence based on the variance of information entropy. However, according to reference [23], there are three drawbacks of this approach. (i) It does not satisfy the property of the additive consistency and the order consistency; (ii) It cannot calculate the preference values in some situations; (iii) The preference values in the consistency matrix are not always between zero and one. Therefore, a new improved DST approach is proposed in this paper inspired by reference [26], which well considers the combination of unreliable evidence in the group decision making under the framework of FPR.

Two major contributions have been made in this work. First, a new hybrid classifier ensemble (HCE) method is proposed based on entropy features to improve the performance and accuracy of fault diagnosis. Second, an improved DST has been proposed to perform information fusion of classification decisions obtained by HCE, which considers the combination of unreliable and conflictive evidence sources, the uncertainty information of basic probability assignment (BPA) and the relative credibility of the evidence on the weights under the framework of FPR. The novel HCE model combined with the improved DST technique has been utilized to automatically identify bearing faults in a rotating machine. Results have demonstrated well the effectiveness of the proposed method.

This work is organized as follows. Theories of entropy feature extraction and single-stage classifier have been briefly reviewed in Section 2. The improved DST for dealing with conflicting evidence has been given in Section 3, where the performance of the proposed approaches has also been demonstrated using two examples. The HCE approach combined with the improved DST is adopted to identify bearing fault automatically, whose effectiveness was demonstrated on a test-rig in Section 4. Conclusions are drawn in Section 5.

## 2. Methodologies

The techniques of entropy feature extraction and the classifiers mentioned in HCE have been briefly introduced in this section. 

### 2.1. Entropy Feature Extraction

Feature extraction is crucial in pattern recognition and mechanical fault diagnosis. However, traditional signal processing methods, like Fourier transform, are not suitable for analyzing the non-linear and non-stationary bearing vibration signals. It seems that time-frequency analysis techniques are much more suitable for extracting bearing fault features. Several advanced time-frequency signal processing techniques have been adopted in feature extraction. For example, variational mode decomposition (VMD) [27] is as a self-adaptive decomposition method lately proposed with a solid theory [28]. 

Moreover, traditional statistical properties and frequency-domain signatures cannot meet the requirements because of the non-linear and non-stationary characteristics of the decomposed components [29]. Many non-linear parameter estimation methods have been proved to get the feature information, such as entropy theory introduced in reference [30] to estimate the complexity and stationarity of the signal. Entropy features can be also applied to quantify the malfunction and reflect the uncertainty of vibration signals. In addition, different entropy features obtained in different domains can be used to fully describe a vibration signal. Thus, singular spectrum entropy (SSE) [31], power spectrum entropy (PSE) [32], time-frequency entropy (TFE) [33], and wavelet packet energy spectrum entropy (WPESE) [34] have been used to calculate the feature sets in this work, which are associated with singular spectrum in time domain, power spectrum in frequency domain, time-frequency spectrum, and wavelet packet energy spectrum in time-frequency domain, respectively. These four entropy features will be indicated as follows.

#### 2.1.1. Singular Spectrum Entropy

SSE indicates the uncertainty degree of the signal energy divided by singular spectrum analysis, which can effectively represent the signal energy change in the time domain [31]. Based on the delay embedding technique, an arbitrary signal $\left\{x_{i}\right\}\left(i=1,\;2,\ldots ,\;N\right)$ was mapped to an embedded space represented by the *M* × *N* matrix U, i.e., As explained in reference [31], the calculation of U is shown as
(1)$$U=\left[\begin{array}{cccc} x_{1} & x_{2} & \ldots & x_{M}\\ x_{2} & x_{3} & \ldots & x_{M+1}\\ \vdots & \vdots & \ldots & \vdots \\ x_{N-M} & x_{N-M+1} & \ldots & x_{N} \end{array}\right]$$
where *M* is the length of the embedded space, *N* is the number of samples. The singular values {*λ_i_*} of the matrix U are achieved based on the singular value decomposition (SVD). Thus, the SSE of the signal via information entropy theory is defined as
(2)$$H_{S}=-\overset{M}{\underset{i=1}{\sum }}p_{i}\log p_{i}$$
in which
(3)$$p_{i}=\lambda_{i}/(\overset{M}{\underset{i=1}{\sum }}\lambda_{i})$$
and $p_{i}$ is the ratio of the ith singular spectrum to the whole spectrum.

#### 2.1.2. Power Spectrum Entropy

PSE can reflect the complexity and stability of a signal, which is also used to indicate the distribution of signal energy in frequency domain [32]. The proportional distribution of different frequencies is defined as a probability distribution. When X(ω) is obtained by using the discrete Fourier transform for a signal$\;\left\{x_{t}\right\}$, as explained in reference [32], the calculation of the power spectrum is shown as
(4)$$S\left(\omega \right)=\frac{1}{N}{\left|X_{i}\left(\omega \right)\right|}^{2}.$$
where $S=\left\{S_{1},S_{2},\ldots ,S_{N}\right\}$ can be regarded as the partition of a signal$\;\left\{x_{t}\right\}$. Hence the PSE can be defined as follows:(5)$$H_{P}=-\overset{N}{\underset{i=1}{\sum }}q_{i}\log q_{i}.$$
where $q_{i}=S_{i}/(\sum_{i=1}^{N}S_{i})$, and $q_{i}$ is the ratio of the *i*th power spectrum to the whole spectrum.

#### 2.1.3. Time-Frequency Entropy

TFE is used to quantitatively measure the time-frequency representation [33]. Let a time-frequency plot have *L* equal blocks, where the information source for the entire plane is η and for each block is$\;\gamma_{i}\left(i=1,2,\ldots ,L\right)$. As explained in reference [33], the calculation of the time-frequency entropy is shown as
(6)$$H_{T}=-\overset{N}{\underset{i=1}{\sum }}\delta_{i}\log\delta_{i}.$$
where $\delta_{i}$ = $\gamma_{i}/\eta$, $\delta_{i}\;$the ratio of the *i*-th energy to the whole energy.

#### 2.1.4. Wavelet Packet Energy Spectrum Entropy

A sequence $\left\{J_{k}^{j},\;k=0,1,2,\ldots ,2^{j}-1\right\}\;$represents the decomposition result using *j*-layer wavelet package transform. The sum of squares of signals in each frequency band after wavelet packet transform (WPT) is selected as wavelet packet energy. As explained in reference [34], the calculation of energy value corresponding to the *i*-th band is given below
(7)$$E_{i}=\overset{2^{j}}{\underset{k=1}{\sum }}{\left|W_{i}\left(k\right)\right|}^{2}.$$
where $W_{i}\left(k\right)\;$is the reconstructed coefficients for each node. Thus, WPESE can be defined by
(8)$$H_{W}=-\overset{2^{j}}{\underset{i=1}{\sum }}r_{i}\log r_{i}\;.$$

### 2.2. Classification Models

The difference between classifiers in HCE should be increased to enhance the complementarity between classification methods, which can comprehensively describe the diagnostic object. Three supervised classification models are selected, that is, the traditional Deep Neural Networks (DNN), the shallow learning algorithm Support Vector Machine (SVM), and Extreme Learning Machine (ELM).

DNN is one of the most widely used intelligent methods in pattern recognition, fault diagnosis and classification. DNN is a kind of deep learning technique, which is comprised of unsupervised layer-by-layer greedy training and global parameter tuning using the back propagation algorithm. DNN can not only solve complex nonlinear problems but also extract features in a high-dimensional space. Presently, many different models of DNN have been developed. For example, a DNN-based model was used to identify the fault condition of roller bearing [35]. The Deep Boltzmann machine combined with multi-grained scanning forest ensemble was developed for the fault diagnosis of industrial big data [7]. Thus, DNN will be adopted as single-stage classifier in HCE in this work.

SVM is a well-known shallow learning method in classification and regression applications. SVM has good generalization capability for classification of a small sample [36], which have been widely used in fault diagnosis and prognostics. To improve the performance of SVM, PSO is adopted to optimize the parameters in SVM. 

ELM is considered as a single hidden layer feed forward neural networks [37,38]. The input weights are set randomly, then the network is expressed as a linear system, and the output weights can be calculated analytically [38]. The weight between the hidden layer and the output layer of ELM does not need to be adjusted iteratively, which is obtained by generalized inverse of a matrix. The performance of ELM depends on the randomly input weights and thresholds. In this work, the fruit fly optimization algorithm (FOA) is used to improve the performance of traditional ELM. Both SVM and ELM are utilized in HCE in this work.

### 2.3. Dempster-Shafer Theory

DST is one of the most powerful tools for the ensemble of multiple classifiers system, which can deal with incomplete, uncertain, and unclear information in the multi-sensor information fusion [39]. DST was initially developed by Shafer in 1976. Assume $\Theta =\left\{D_{1},D_{2},\ldots ,D_{n}\right\}$ is a set of mutually exclusive and collectively exhaustive events, which is called the frame of discernment (FOD). A basic probability assignment (BPA) is a map of m from$\;2^{\Theta }\;$to [0, 1], as explained in reference [40], the calculation of the BPA function is shown below,
(9)$$\{\begin{array}{c} \sum_{A\subset 2^{\Theta }}m\left(A\right)=1\\ m\left(\emptyset \right)=0 \end{array}.$$

Based on the belief function theory, two independent BPAs can be combined by Dempster’s rule, denoted as$\;m=m_{1}⊕m_{2}$, which is defined as follows.
(10)$$m\left(A\right)=\{\begin{array}{cc} \frac{1}{1-K}\sum_{B\cap C=A}m_{1}\left(B\right)m_{2}\left(C\right),\; & A\neq \emptyset \\ 0, & A=\emptyset \end{array}$$
where $K=\sum_{B\cap C=\emptyset }m_{1}\left(B\right)m_{2}\left(C\right).$ The conflict coefficient *K* is used to measure the conflict between two pieces of evidences. The larger the value of *K* is, the larger conflict between evidences gets. 

It should be noted that there may exist conflict between the evidence in the fusion of HCE. To solve this issue, a new weighted average approach is proposed, which considers not only the support degree between the pieces of evidence but also the uncertainty information of BPA. This improved version of DST is given in the following subsection.

## 3. The Improved Dempster-Shafer Theory Approach

It is crucial to detect the relatively reliable evidence in the process of information fusion. In the multiple classifier systems, the conflict problem caused by the result of the classifier cannot be ignored. Thus, an improved DST approach is developed in this work and will be introduced in detail subsequently. First, since cosine similarity reflects the confidence degree of the evidence itself, the cosine similarity is employed to indicate the support degree between the pieces of evidence. In addition, DST can be considered as a generalized probability theory, entropy can be used to measure the quantitative uncertainty in BPA. Therefore, entropy based on FPR is applied to indicate the relative reliability preference between the bodies of evidence (BOE). Considering the above two aspects, it can be found that the improved DST will be much more reasonable in dealing with conflicts compared with the original DST. The proposed technique includes three parts: The measurement of the degree of support between evidence using the cosine similarity, the calculation of the weight of BPA, and the improved fusion for BPAs, as shown in Figure 1. 

### 3.1. The Cosine Similarity

The cosine similarity is used to measure the confidence degree of evidence [41]. Let Θ be a frame of discernment and$\;\Theta =\left\{\theta_{1},\theta_{2},\ldots ,\theta_{n}\right\}.\;$Employ the cosine similarity function, as explained in reference [41], the calculation of similarity degree between evidence$\;m_{i},\;m_{j}$ is given below,
(11)$$S_{ij}=\frac{m_{i}\cdot m_{j}^{T}}{∥m_{i}∥\cdot ∥m_{j}∥}.$$
where $m_{i}\cdot m_{j\;}$is inner product of$\;m_{i}\;$and$\;m_{j}$. And ∥⋅∥ represents the norm of vector. For the *n*-sources fusion system, the similarity measure matrix is defined as follow.
(12)$$S=\left[\begin{array}{ccccc} 1 & \cdots & S_{1i} & \cdots & S_{1k}\\ \vdots & \vdots & \vdots & \vdots & \vdots \\ S_{i1} & \cdots & 1 & \cdots & S_{ik}\\ \vdots & \vdots & \vdots & \vdots & \vdots \\ S_{k1} & \cdots & S_{ki} & \cdots & 1 \end{array}\right]\;.$$

The Support degree of the evidence$\;m_{i}\;$can be defined as follows.
(13)$$sup(m_{i})=\overset{n}{\underset{j=1}{\sum }}S_{ij}\;.$$

Thus, the credibility degree of the evidence$\;m_{i}\;$is denoted below.
(14)$$crd_{i}=\frac{sup(m_{i})}{\max\left(sup(m_{i}\right))}\;.$$

### 3.2. The Uncertainty Measurement of the Weights

Deng entropy [42], which is used to measure the quantitative uncertainty of BPA in this work. Assume m(⋅) is a mass function defined on the frame of discernment, as explained in reference [42], the calculation of Deng entropy$\;E_{d}\left(m\right)\;$of the BPA is shown as
(15)$$E_{d}\left(m\right)=-\underset{A\subseteq \Theta }{\sum }m\left(A\right)\log_{2}\frac{m\left(A\right)}{2^{\left|A\right|}-1}.$$
where *A* is the focal element of *m*, |A| is the cardinality of *A*.

The FPR analysis based on the Deng entropy is adopted to denote the relative reliability preference between bodies of evidence. Fuzzy sets have been widely used in various applications and play an important role in the decision-making process [43]. The concepts of FPR and the additive consistency of FPR are introduced briefly.

The fuzzy preference matrix is construct by the variance of entropy. If the system has more than two pieces of evidence, as explained in reference [25], the calculation of variance of entropy is shown as
(16)$$V_{i}=e^{E_{d}\left(m_{i}\right)},\;1\leq i\leq k$$
(17)$${Var}_{i}=\mathrm{Var}\left(\left\{{\bar{V}}_{1},{\bar{V}}_{2},\ldots ,{\bar{V}}_{i-1},{\bar{V}}_{i+1},\ldots ,{\bar{V}}_{k}\right\}\right).$$
where ${\bar{V}}_{i}=V_{i}/\sum_{i=1}^{k}V_{i}$, and ${Var}_{i}$ denotes the variance. Then, the off-diagonal elements $\rho_{ij}\;$and $\rho_{ji}\;$of the fuzzy preference matrix can be computed by.
(18)$$\rho_{ij}=\frac{Var_{i}}{Var_{i}+Var_{j}},\;\rho_{ji}=\frac{Var_{j}}{Var_{i}+Var_{j}}\;.$$

Let P be a fuzzy preference matrix for the set M of alternatives $M=\{M_{1},M_{2},\ldots M_{n}\}$, as explained in reference [43], the defined of P is shown as
(19)$$P={\left(\rho_{ij}\right)}_{n\times n}=\left[\begin{array}{ccc} 0.5 & \begin{array}{cc} \rho_{12} & \cdots \end{array} & \rho_{1n}\\ \begin{array}{c} \rho_{21}\\ \vdots \end{array} & \begin{array}{cc} \begin{array}{c} 0.5\\ \vdots \end{array} & \begin{array}{c} \cdots \\ \ddots \end{array} \end{array} & \begin{array}{c} \rho_{2n}\\ \vdots \end{array}\\ \rho_{n1} & \begin{array}{cc} \rho_{n2} & \cdots \end{array} & 0.5 \end{array}\right].$$
where $\rho_{ij}\;$denotes the degree of preference of alternative $M_{i}\;$over alternative $M_{j}$. Let *P* be a fuzzy preference relation $P={\left(\rho_{ij}\right)}_{n\times n}$, if *P* is a complete FPR as explained in reference [44], which satisfies the following additive consistency properties for all *i*, *j* and *k*.
(20)$$\{\begin{array}{c} \rho_{ij}+\rho_{ji}=1,\\ \rho_{ii}=0.5,\\ P_{ik}=P_{ij}+P_{jk}-0.5 \end{array}.$$
where 1≤i≤n, 1≤j≤n and 1≤k≤n, then *P* is called an additive consistent FPR. Based on the complete fuzzy preference relation *P*, as explained in reference [26], a consistency matrix $\bar{P}$ which satisfies the additive consistency is shown as
(21)$$\bar{P}={\left({\bar{\rho }}_{ik}\right)}_{n\times n}={\left(\frac{1}{2n}\overset{n}{\underset{j=1}{\sum }}\left(\rho_{ij}-\rho_{ji}+\rho_{jk}-\rho_{kj}\right)+0.5\right)}_{n\times n}.$$

And then, as explained in reference [26], the calculation of the boundary constant ξ and the consistency degree ς are shown as
(22)$$\{\begin{array}{c} \chi_{i}=\frac{1}{n}\sum_{j=1}^{n}{\bar{\rho }}_{ij}\\ \varepsilon =\max\left(\chi_{i}|1\leq i\leq n\right)\\ \begin{array}{c} \mu =\min\left(\chi_{i}|1\leq i\leq n\right)\\ \xi =\frac{1}{2\cdot \max\left(0.5,\;\left(\varepsilon -\mu \right)\right)}\\ \varsigma =1-\frac{2}{n\left(n-1\right)}\sum_{i=1}^{n}\sum_{k=1,k\neq i}^{n}\left|\rho_{ik}-{\bar{\rho }}_{ik}\right| \end{array} \end{array}\;.$$
where $\chi_{i}\;$is the average value of preference values of alternative, ε is the maximum value of all $\chi_{i}$, μ is the minimum value of all$\;\chi_{i}$, ξ is the boundary constant to let the preference values in the consistency matrix $\bar{P}$ is between zero and one, ς represents the consistency degree between P and $\bar{P}$. The larger the value of ς, the more the consistency of the fuzzy preference relation. If the value of ς is close to one, then the information of fuzzy preference relation is more consistent ξ∈[0,1], ς∈[0,1], 1≤i≤n, 1≤k≤n. As explained in reference [26], the calculation of the modified consistency matrix $\tilde{P}$ is shown as
(23)$$\tilde{P}={\left({\tilde{\rho }}_{ik}\right)}_{n\times n}={\left({\bar{\rho }}_{ik}\times \kappa +\frac{1}{2}\left(1-\kappa \right)\right)}_{n\times n}.$$
where κ denotes the modified constant. And κ=ξ×ζ, κ∈[0,1]. The ranking value $R_{i}\;$of the alternative$\;M_{i}\;$in the set M is calculation as follows
(24)$$R_{i}=\frac{2}{n^{2}-n}\overset{n}{\underset{j=1,j\neq i}{\sum }}{\tilde{\rho }}_{ij}\;.$$
where 1≤i≤n, 1≤j≤n and$\;\sum_{i=1}^{n}R_{i}=1$.

### 3.3. The Improved Fusion Algorithm

With the credibility degree$\;crd_{i}\;$and the ranking value of alternative BPAs$\;R_{i}$, the support degree of the BPA is denoted as$\;P_{Sup_{i}}$,
(25)$$P_{Sup_{i}}=crd_{i}\times R_{i}\;.$$

Based on the weight$\;P_{Sup_{i}}$, the weighted average of the evidence (WAE) is given as follow.
(26)$$\mathrm{WAE}\left(m\right)=\overset{k}{\underset{i=1}{\sum }}\left({\bar{P}}_{Sup_{i}}\times m_{i}\right).\;$$
where$\;{\bar{P}}_{Sup_{i}}=P_{Sup_{i}}/\sum_{i=1}^{k}P_{Sup_{i}}$. Therefore, the modified mass function obtained by Equation (26) will be fused with Dempster’s rule of combination *n*-1 times when there are *n* pieces of evidence.

### 3.4. Numerical Verification

A numerical example obtained from reference [21] is illustrated to verify the effectiveness of the improved method in dealing with conflict evidences. Suppose the recognition target is A based on multiple sensor data given in Table 1. It showed five different types of sensors, and the FOD is given by Θ={A,B,C}. The results using different combination rules are shown in Table 2.

As can be seen in Table 2, although more evidence supports target *A*, a wrong decision was still achieved with Dempster’s method. When the number of evidence were not adequate, the performance of Murphy’s method was not satisfactory. Obviously, the simple averaging and other weight averaging can provide reasonable results, but the proposed method in this work is much better in dealing with conflicting evidence.

### 3.5. An Example of Fault Diagnosis Application

Another example given in reference [45] has been utilized to further demonstrate the effectiveness of the improved DST in fault diagnosis. The BPAs of the sensor data are directly adopted from reference [46]. Suppose the frame of discernment is F, which have three types of fault in a motor rotor, denoted as$\;F_{1}=\left\{Rotor\;unbalance\right\}$,$\;F_{2}=\left\{Rotor\;misalignment\right\},\;$and $F_{3}=\left\{Pedestal\;looseness\right\}$, respectively. Three vibration accelerometer sensors are installed in different positions to collect the vibration signals, denoted by$\;S=\left\{S_{1},S_{2},S_{3}\right\}$. The frequency of vibration signal locating at 1×, 2× and 3× (× denotes rotor rotating frequency) are considered as the fault features, as are shown in Table 3.

The modified mass function could also be calculated with the proposed method. The weighted average of the evidence shown in the Table 4 can be obtained by Equation (26). It can be seen that the probability of $F_{2}$ is the largest, which can be preliminarily judged as the fault type. The modified mass function will be fused with Dempster’s rule of combination. The fusion results given in reference [46] were obtained by Equation (10) using the Dempster’s rule 2 times, which is also shown in Table 5, Table 6 and Table 7. The corresponding Target column represents the fault type for fusion diagnosis.

The improved DST is used to solve the fusion issue in the fault diagnosis mentioned above. According to the results shown in Table 5, Table 6 and Table 7, the conflict of sensor reports has been solved with the proposed method. We can notice that the proposed method can successfully detect the fault type$\;F_{2}$, which is consistent with those given in reference [46]. Thus, both the two methods can conduct the conflictive pieces of evidence and identify the fault type$\;F_{2}$ well. Moreover, it can be seen in Figure 2, Figure 3 and Figure 4 that the proposed method can deal well with the conflictive pieces of evidence. The belief degrees assigned to the target $F_{2}$ at 1× frequency, 2× frequency and 3× frequency using the proposed method were separately 0.9277, 0.9858, and 0.6321, which are all higher than the method in reference [46].

## 4. Experimental Analysis

The effectiveness of the improved Dempster-Shafer (D-S) evidence theory in dealing with conflicting evidence has been verified in the previous section. The proposed HCE framework in roller bearing fault diagnosis and the robustness of improved DST in information fusion will be illustrated in this section. The present technique is then applied for the rolling bearing fault diagnosis experiments on the Machinery Fault Simulator Magnum (MFS-MG) test-rig. The flowchart of the fault diagnosis using the proposed procedure is shown as Figure 5.

### 4.1. The Experimental Set-Up

As shown in Figure 6, the vibration data set were acquired on the MFS-MG test rig, and the defective bearing of the type ER-12K was installed on the left side of the shaft. Accelerometer sensors were installed in vertical and horizontal on bearing seats. Sampling frequency was set to 25,600 Hz, and the rotating frequency of the motor was 29.87 Hz (about 1792 rpm). The fault types: Ball (B), cage (C), inner race (IR) and outer race (OR), as well as a normal (N) condition were used in the experiments. Each segment of the collected original vibration signal had 10,240 data points. The original vibration data and their frequency spectra are shown in Figure 7.

### 4.2. Entropy Feature Sets

We could obtain four entropy features, the features of vibration signals. The original vibration signal was decomposed with the VMD method, and the decomposed intrinsic mode function (IMF) were achieved. The key parameters used in VMD should be selected based on the empirical value, interested readers can refer to reference [47]. Assume$\;IMF_{i}=\left\{x_{1},x_{2},\ldots ,x_{K}\right\}$, where *K* is the number of data points of$\;IMF_{i}$. The SSE, PSE, and TFE of each$\;IMF_{i}\;$were extracted using Equations (2), (5), and (6), respectively. Moreover, the WPESE of each original segment was also obtained using Equation (8). Here, a 3-level decomposition was used in WPT with the selected mother wavelet Db10. Since there were 112 samples for each experimental condition, the numbers of rows and columns in the feature matrix were 560 and 4, respectively. Figure 8 shows the entropy feature sets. The datasets were divided into two parts, and the former 75% of each class of data was randomly selected as training data, while the remaining 25% was testing data. The training data and the testing data was defined as a 420(row)–5(column) matrix and a 140(row)–5(column) matrix, respectively. The desirable classes were labeled with 1, 2, 3, 4, and 5. For example, outputs 1 and 3 were separately related to the first and the third class. In this way, three supervised classifiers could be used to identify the bearing faults.

### 4.3. Classification Using Single-Stage Classifier

DNN, SVM, and ELM were separately adopted in the single-stage classification based on the above achieved entropy signatures. In this work, a large number of neurons were tested to find an optimal structure of DNN. The number of hidden layer neurons which resulted in the highest classification accuracy was selected as the optimum number. Then, the optimum DNN structure was constructed based on the obtained number of hidden layer neurons. Figure 9 shows the classification accuracies of DNN based on the different numbers of hidden layer neurons and mini-batch gradient descent (MBGD) algorithm. It can be seen in Figure 10 that the determined optimal number of hidden layer neurons is set to 13. 

In the SVM technique, the Gaussian radial basis function (RBF) was selected as the kernel function, and the particle swarm optimization (PSO) was used to determine the optimized parameters in the SVM. The population size (pop), maximum number of iterations (maxgen), two acceleration constants ($c_{1},c_{2}$), and the inertia weight (ψ) were set to$\;c_{1}=1.5,\;c_{2}=1.7\;$and ψ=1, pop = 20, maxgen = 100, respectively. In addition, the parameters of FOA used in ELM, such as the population size (pop) and maximum number of iterations (maxgen) were set to 20, 100, while the initial positions were set randomly.

After data training using each classifier, the testing data set was used to validate the accuracy of each classifier model for bearing fault diagnosis. The aim of classification was to assign an input pattern to one of the 5 classes concerned in the present study and represented by the classification labels. The classification results of the testing data set obtained by preliminary diagnosis are shown in Figure 10, Figure 11 and Figure 12. The performances of DNN, ELM, and SVM are illustrated in Table 8, Table 9 and Table 10, respectively. The meaning of Y-axis in Figure 10a, Figure 11a, and Figure 12a represents five bearing conditions, denoted by four fault types B, C, IR, OR as well as a normal condition (N).

Figure 10a shows the desired output and the output of the trained DNN. Figure 10b shows the absolute error of the DNN output with respect to the desired output, where a sample is misclassified when the absolute error is large. As can be seen from Table 8, the average classification accuracy of DNN is 88.57%. Figure 11a illustrates the desired output and the output of the trained ELM, while Figure 11b shows the absolute error of the ELM output with respect to the desired output. As can be seen from Table 9, the average classification accuracy of the testing data set using the ELM approach is about 80.81%. Similarly, Figure 12a shows the desired output and the output of the trained SVM, and Figure 12b shows the absolute error of the SVM output with respect to the desired output. As can be seen from Table 10, the average classification accuracy of the testing data set using the SVM approach is only 77.14%.

It can be found that the classification rates separately using these three techniques were not good enough. Among them, DNN achieved the best classification results based on the deep learning technique as well as its optimal structures, compared with SVM and the ELM. The accuracy using single-stage classifier was still not good enough. Therefore, the data fusion method is necessary to be employed to increase the classification accuracy.

### 4.4. Results Using the HCE Algorithm and the Improved DST

Since the classification results were separately obtained using a single classifier, their results can be syncretized further. In this work, the fusion of the primary classification results was carried out using the improved DST method. First, three types of evidence were introduced as follows. $E_{1},\;E_{2}$, and $E_{3}$ were the classification results using the supervised classifiers DNN, ELM, and SVM, respectively. The original Dempster’s rule and the proposed method were both used to achieve the fusion results. In fact, the counter-intuitive results are often obtained when Dempster’s rule of combination is utilized in some cases, especially, when the BOEs to be combined are highly conflicting. 

In order to improve the diagnostic accuracy, DST and the proposed DST were used to fuse the preliminary diagnosis of HCE. The results of different methods are given in Table 11. In the fusion stage, each testing sample corresponded to a probabilistic output, which was the body of evidence. The meaning of X-axis in Figure 13, Figure 14 and Figure 15 represents 140 bodies of evidence, while the meaning of Y-axis in Figure 13, Figure 14 and Figure 15 represents fusion results of evidence using different methods. The fusion result of HCE by the proposed DST is shown in Figure 13, while the fusion result using HCE and the original DST is shown in Figure 14. A sample is misclassified when its fusion result is smaller than or equal to 0.5. It can be seen in Figure 13 and Figure 14 that the classification accuracy using the proposed HCE and the improved DST is the highest, about 97.86%. In addition, the accuracy using the original DST is about 92.86%, which is also better than those using a single-stage classifier. Figure 15 illustrates the results using the technique given in reference [25]. We can find the result is better than those achieved using original DST, but it is still worse compared with our proposed methods. This well demonstrated that the proposed HCE approach combined with the improved DST can reliably be automatically used for roller bearing fault detection. It means that the fault detection accuracy can significantly be improved by applying HCE approach. 

## 5. Conclusions

It is crucial to detect the relatively reliable evidence with the collected multi-source evidence in the process of information fusion. The HCE approach combined with the improved DST has been proposed for the fault diagnosis of roller bearings. The effects of support degree among the pieces of evidence, the uncertainty information of BPA, and the relative credibility of the evidence on the weights are all considered in this improved DST. The improved DST can effectively deal with conflicts between the evidences and then improve the diagnostic accuracy. The cosine similarity is employed to indicate the confidence degree between the pieces of evidence. Entropy features are used to measure the quantitative uncertainty of BPA in the improved DST. In addition, entropy based FPR is employed to indicate the relative reliability preference between BOEs. Thus, the improved DST is much more reasonable in dealing with conflicts compared with the original DST. The effectiveness of the improved Dempster-Shafer theory has been verified via two examples. 

In addition, SSE, PSE, TFE, and WPESE features have been utilized in the single-stage classification with DNN, SVM, and ELM in this work. Performances of the proposed HCE approach combined with the improved DST has been demonstrated on a bearing test-rig, compared with the original DST. It can be found that the overall error rate of the HCE approach can be greatly reduced using the improved DST, while the accuracy of the rolling element bearings diagnosis is successfully raised. Since there is not enough (complete) fault data for a rotating machine in practice, it is usually difficult dealing with a small sample and incomplete data in the process of decision-making. The proposed technique will be further investigated under these cases in the future.

## Figures and Tables

**Figure 1 sensors-19-02097-f001:**
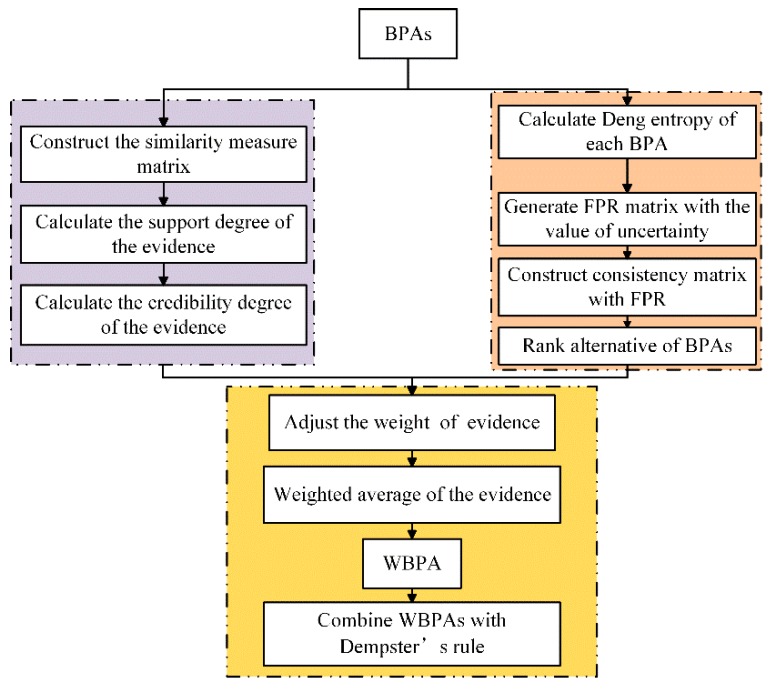
The flowchart of the proposed Dempster-Shafer theory (DST).

**Figure 2 sensors-19-02097-f002:**
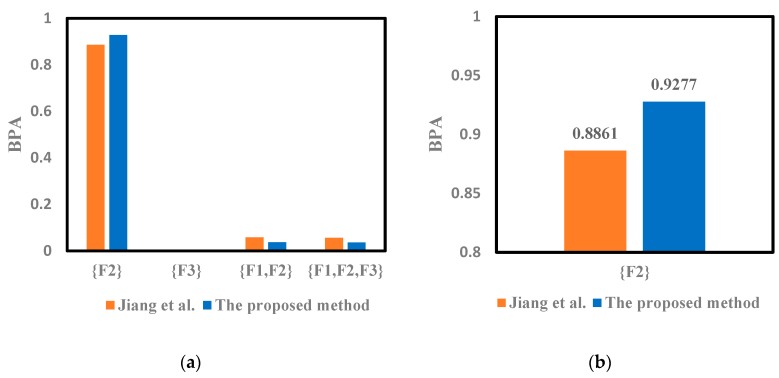
The comparison of different methods for motor rotor fault diagnosis at 1X frequency. (**a**) Fusion results of different methods. (**b**) The result of {$F_{2}$} for 1X.

**Figure 3 sensors-19-02097-f003:**
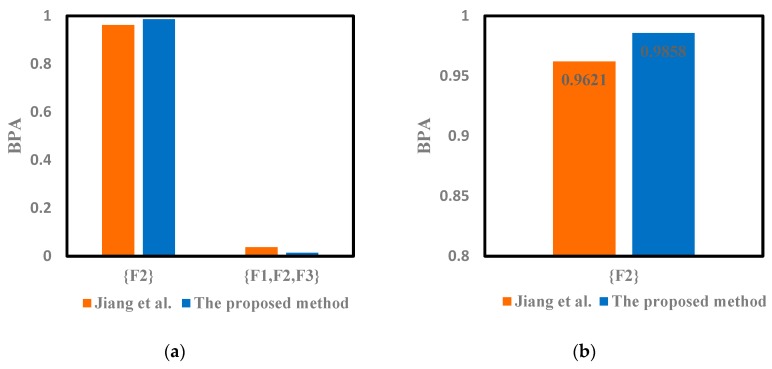
The comparison of different methods for motor rotor fault diagnosis at 2X frequency. (**a**) Fusion results of different methods. (**b**) The result of {$F_{2}$} for 2X.

**Figure 4 sensors-19-02097-f004:**
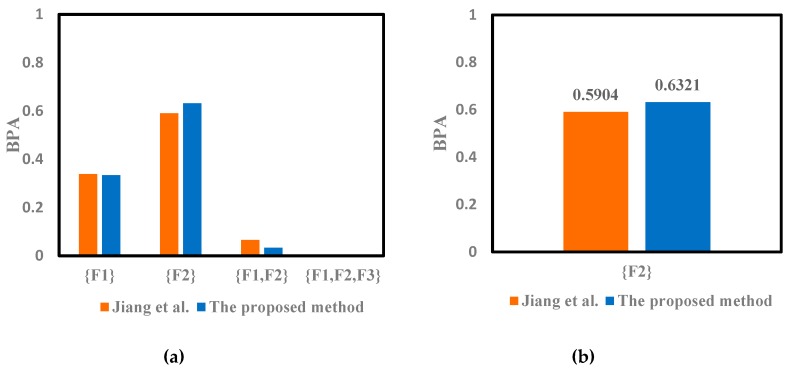
The comparison of different methods for motor rotor fault diagnosis at 3X frequency. (**a**) Fusion results of different methods. (**b**) The result of {$F_{2}$} for 3X.

**Figure 5 sensors-19-02097-f005:**
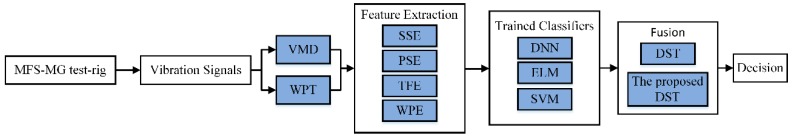
Flowchart of the proposed procedure.

**Figure 6 sensors-19-02097-f006:**
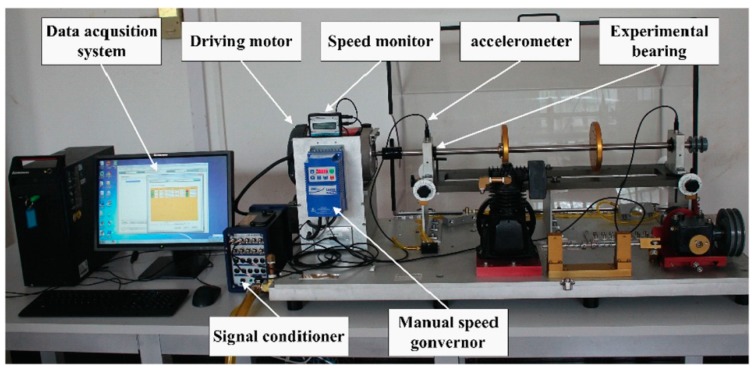
Machinery Fault Simulator Magnum (MFS-MG) fault simulation test bench.

**Figure 7 sensors-19-02097-f007:**
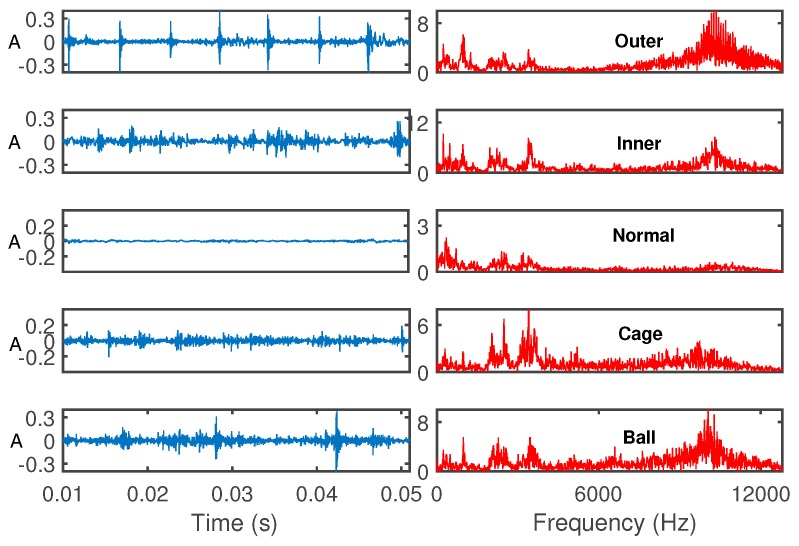
Original vibration signals and their spectra.

**Figure 8 sensors-19-02097-f008:**
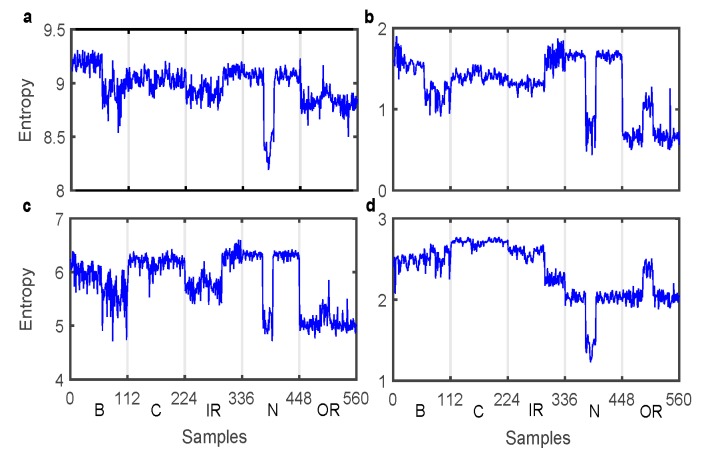
Four kinds of entropy features. (**a**) SSE (**b**) PSE (**c**) TFE (**d**) WPESE.

**Figure 9 sensors-19-02097-f009:**
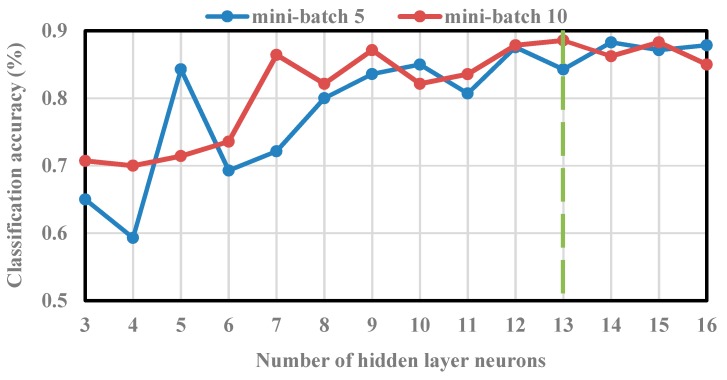
Classification accuracy using deep neural networks (DNN).

**Figure 10 sensors-19-02097-f010:**
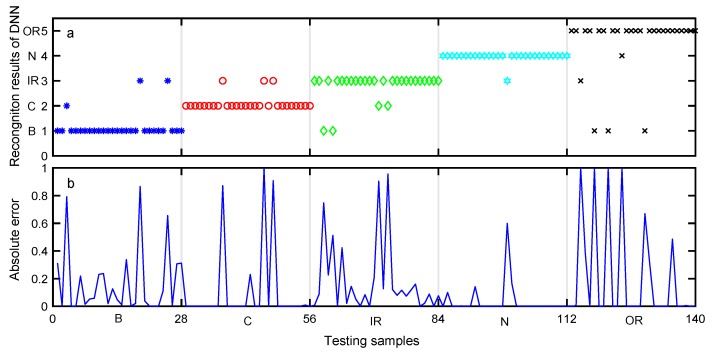
Preliminary diagnosis of DNN (**a**) Recognition results. (**b**) Absolute error of the proposed approach output with respect to the desired output.

**Figure 11 sensors-19-02097-f011:**
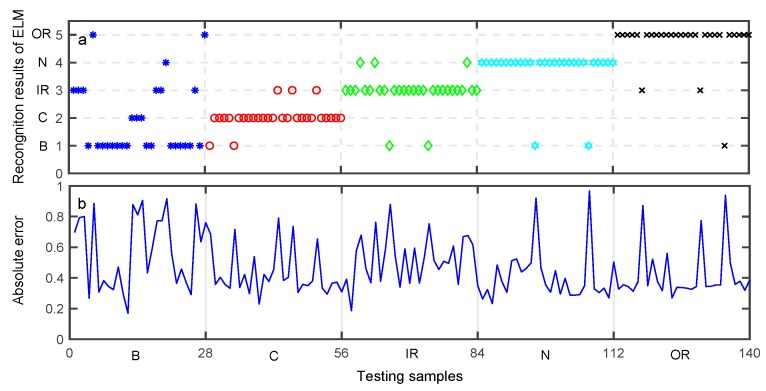
Preliminary diagnosis of extreme learning machine (ELM). (**a**) Recognition results. (**b**) Absolute error of the proposed approach output with respect to the desired output.

**Figure 12 sensors-19-02097-f012:**
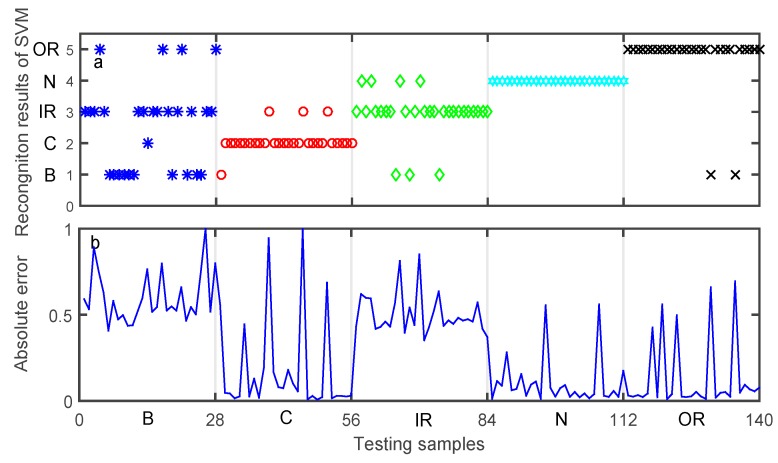
Preliminary diagnosis of support vector machine (SVM). (**a**) Recognition results. (**b**) Absolute error of the proposed approach output with respect to the desired output.

**Figure 13 sensors-19-02097-f013:**
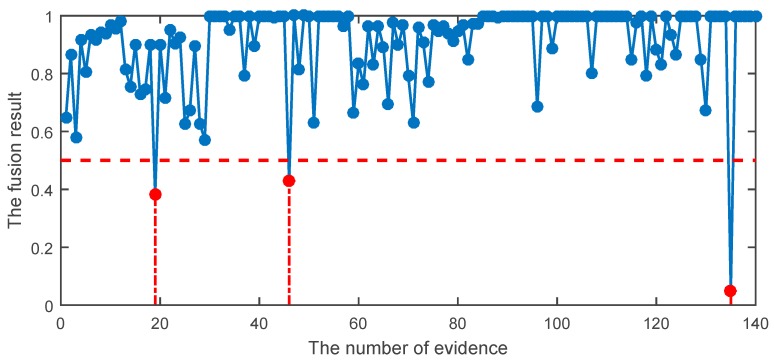
The fusion result using the proposed hybrid classifier ensemble (HCE) combined with the improved DST.

**Figure 14 sensors-19-02097-f014:**
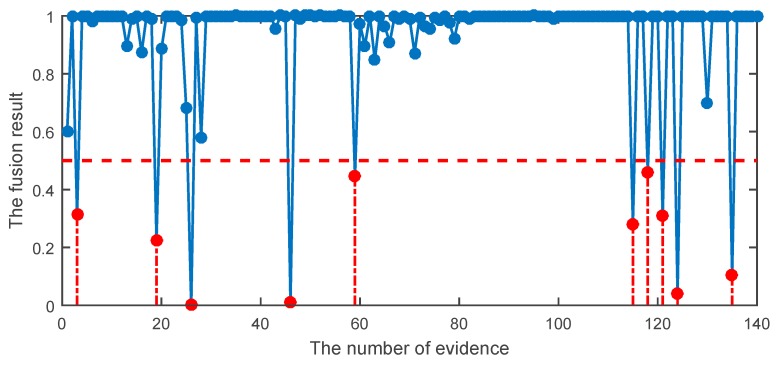
The fusion result using HCE with original DST.

**Figure 15 sensors-19-02097-f015:**
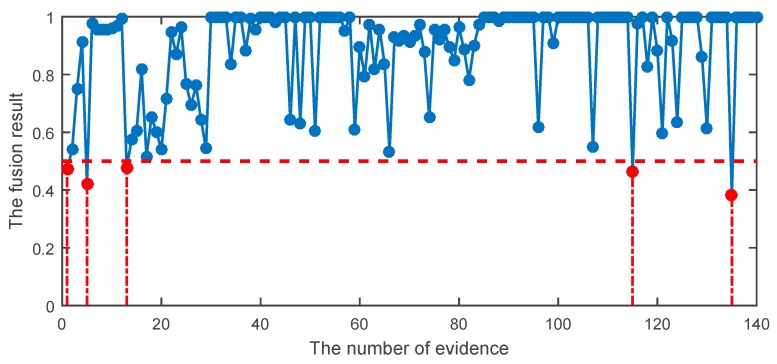
The fusion result using HCE with DST given in reference [25].

**Table 1 sensors-19-02097-t001:** Basic probability assignment (BPA) of the sensor data.

BPA	{*A*}	{*B*}	{*C*}	{*A*,*C*}
$m_{1}$	0.41	0.29	0.30	0.00
$m_{2}$	0.00	0.90	0.10	0.00
$m_{3}$	0.58	0.07	0.00	0.35
$m_{4}$	0.55	0.10	0.00	0.35
$m_{5}$	0.60	0.10	0.00	0.30

**Table 2 sensors-19-02097-t002:** Results of the evidence using different fusion methods.

Evidence	Method	{*A*}	{*B*}	{*C*}	{*A*,*C*}
$m_{1},m_{2},m_{3}$	Dempster	0	0.6350	0.3650	0
	Murphy [20]	0.4939	0.4180	0.0792	0.0090
	Deng et al. [21]	0.4974	0.4054	0.0888	0.0084
	Zhang et al. [22]	0.5681	0.3319	0.0929	0.0084
	The proposed method	0.8308	0.0532	0.1046	0.0115
$m_{1},m_{2},m_{3},m_{4}$	Dempster	0	0.3321	0.6679	0
	Murphy [20]	0.8362	0.1147	0.0410	0.0081
	Deng et al. [21]	0.9089	0.0444	0.0379	0.0089
	Zhang et al. [22]	0.9142	0.0395	0.0399	0.0083
	The proposed method	0.9535	0.0046	0.0334	0.0085
$m_{1},m_{2},m_{3},m_{4},m_{5}$	Dempster	0	0.1422	0.8578	0
	Murphy [20]	0.9620	0.0210	0.0138	0.0032
	Deng et al. [21]	0.9820	0.0039	0.0107	0.0034
	Zhang et al. [22]	0.9820	0.0034	0.0115	0.0032
	The proposed method	0.9886	0.0004	0.0091	0.0032

**Table 3 sensors-19-02097-t003:** The obtained BPAs.

	*Freq*1				*Freq*2		*Freq*3			
	$\left\{F_{2}\right\}$	$\left\{F_{3}\right\}$	$\left\{F_{1},F_{2}\right\}$	$\left\{F_{1},F_{2},F_{3}\right\}$	$\left\{F_{2}\right\}$	$\left\{F_{1},F_{2},F_{3}\right\}$	$\left\{F_{1}\right\}$	$\left\{F_{2}\right\}$	$\left\{F_{1},F_{2}\right\}$	$\left\{F_{1},F_{2},F_{3}\right\}$
$S_{1}:m_{1}$	0.8176	0.0003	0.1553	0.0268	0.6229	0.3771	0.3666	0.4563	0.1185	0.0586
$S_{2}:m_{2}$	0.5658	0.0009	0.0646	0.3687	0.7660	0.2341	0.2793	0.4151	0.2652	0.0404
$S_{3}:m_{3}$	0.2403	0.0004	0.0141	0.7452	0.8598	0.1402	0.2897	0.4331	0.2470	0.0302

**Table 4 sensors-19-02097-t004:** The modified BPAs.

	*Freq*1				*Freq*2		*Freq*3			
	$\left\{F_{2}\right\}$	$\left\{F_{3}\right\}$	$\left\{F_{1},F_{2}\right\}$	$\left\{F_{1},F_{2},F_{3}\right\}$	$\left\{F_{2}\right\}$	$\left\{F_{1},F_{2},F_{3}\right\}$	$\left\{F_{1}\right\}$	$\left\{F_{2}\right\}$	$\left\{F_{1},F_{2}\right\}$	$\left\{F_{1},F_{2},F_{3}\right\}$
$m_{W}$	0.5836	0.0006	0.0870	0.3288	0.7576	0.2424	0.3109	0.4345	0.2118	0.0428

**Table 5 sensors-19-02097-t005:** Fusion results of different methods for motor rotor fault diagnosis at 1X frequency.

Method	$\left\{F_{2}\right\}$	$\left\{F_{3}\right\}$	$\left\{F_{1},F_{2}\right\}$	$\left\{F_{1},F_{2},F_{3}\right\}$	Target
Jiang et al. [46]	0.8861	0.0002	0.0582	0.0555	$F_{2}$
The proposed method	0.9277	0.0002	0.0364	0.0356	$F_{2}$

**Table 6 sensors-19-02097-t006:** Fusion results of different methods for motor rotor fault diagnosis at 2X frequency.

Method	$\left\{F_{2}\right\}$	$\left\{F_{1},F_{2},F_{3}\right\}$	Target
Jiang et al. [46]	0.9621	0.0371	$F_{2}$
The proposed method	0.9858	0.0142	$F_{2}$

**Table 7 sensors-19-02097-t007:** Fusion results of different methods for motor rotor fault diagnosis at 3X frequency.

Method	$\left\{F_{1}\right\}$	$\left\{F_{2}\right\}$	$\left\{F_{1},F_{2}\right\}$	$\left\{F_{1},F_{2},F_{3}\right\}$	Target
Jiang et al. [46]	0.3384	0.5904	0.0651	0.0061	$F_{2}$
The proposed method	0.3343	0.6321	0.0334	0.0002	$F_{2}$

**Table 8 sensors-19-02097-t008:** Classification accuracy of DNN (%).

Bearing Condition	B	C	IR	N	OR	Average
B	89.29	3.57	7.14	0	0	88.57
C	0	89.29	10.71	0	0	
IR	7.14	7.14	85.71	0	0	
N	0	0	3.57	96.43	0	
OR	10.71	0	3.57	3.57	82.14	

**Table 9 sensors-19-02097-t009:** Classification accuracy of ELM (%).

Bearing Condition	B	C	IR	N	OR	Average
B	57.14	10.71	41.43	3.57	7.14	80.81
C	7.14	82.14	10.71	0	0	
IR	7.14	0	82.14	10.71	0	
N	7.14	0	0	92.86	0	
OR	3.57	0	7.14	0	89.29	

**Table 10 sensors-19-02097-t010:** Classification accuracy of SVM (%).

Bearing Condition	B	C	IR	N	OR	Average
B	25	3.57	53.57	0	14.29	77.14
C	3.57	85.71	10.71	0	0	
IR	10.71	0	75	14.29	0	
N	0	0	0	100	0	
OR	7.14	0	0	0	92.86	

**Table 11 sensors-19-02097-t011:** Results of classification methods.

Method	Classification Rate (%)
HCE with improved DST	97.86
HCE with DST in [25]	96.43
HCE with DST	92.86
DNN	88.57
SVM	77.14
ELM	80.81

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
