# Peer review of "Bearing Fault Diagnosis Based on a Hybrid Classifier Ensemble Approach and the Improved Dempster-Shafer Theory"

_sensors, 2019, doi:10.3390/s19092097_

Round 1

Reviewer 1 Report

Summary

This paper presents an approach for information fusion for Bearing fault diagnosis. The area of potential interest to a wide audience. The approach is the combine ensemble classified with improved Dempster Shafer Theory.  Numerical analysis and experimental work is provided.

While a definite body of work is being presented, the work needs re-shaping before it can be assessed further for review.  In summary:

-          Fuller explanation of the context of the work – the purpose of the methodology/ experimental work;

-          Clearer explanation of the originality of the work  relative to the state of the art;

-          Clearer explanation as to the justification of the selected configurations of DNN/ SVM were used ;

-          More extensive explanation of results;

-          Less reproduction of underlying theory and mathematics for well known concepts;

Detailed comments are supplied here which will help on make the work clearer and more usable by other researchers:

Introduction

The introduction should provide more explanation on the physical context of the problem being solved. Roller bearings are mentioned.   What is the typical “single information source” problem in paragraph 1.  Is it that there is a single sensor/ piece of hardware to provide information? If the new HCE algorithm can now fuse information, what are the multiple information sources?

Second paragraph “It can also be found that Dempster’s combination rule plays a crucial role in DST”.   The combination rule is  an existing crucial element of DST so it undermines the article to say that this can be found out.  

The “lack of robustness” comment re DST – is this referring to the problem of highly conflicting evidence sources?  Or is t something else?

The “many methods” to manage conflicting evidence- these are all variations of DST (Murphy’s etc) so this should be mentioned. They are not new methods separate from DST

“Evidence distance as a critical factor” should be explained in paragraph 2. Do you mean distance of evidence sources from each other?

The final sentence “These methods can reduce the influence of unreliable evidence sources..”.  This is an important one, but unclear.  Evidence sources (e.g. a sensor prone to failure) can be unreliable.  And evidence itself can be fuzzy (the or ambiguous or uncertain (e.g. a location sensor with error rates). It should be explained more.

Contribution paragraph “Two major contributions”.. The first contribution is a “novel hybrid classifier” – explain here how it is novel?  The second contribution is around DST that deals with conflict and uncertainty. There have been various DST variations that do both of these.  How is their different and better?

The use of a deep learning NN, combined with SVM and DST seems a little random, or at least unexplained.  Why that configuration? 

Methodology

The first part of the methodology should explain first what the purpose and context of the experimental work (“the why”) before going into how it is being done.   It goes straight into detail of features etc without explaining what a “method “ is needed for.  

The various sections on feature extraction – are these typical features used for this type of problem? Why these four sets?  Explain.

2.1. – Entropy feature extraction – “feature extraction is very important in fault diagnosis…” – explain this further.  Usually, features are used in order to identify either machine learning inputs. Is this why?  Explain also the context for signal processing here.

The various section SSE/ PSE etc that have equations shown – Presumably these equations are from the original published source  - make this clearer. These are not contributions here.

There is a full paragraph duplicated (in 2.1.1) with paragraph 2.1

“The four entropy features… have been achieved in this work”… Presumably this means they are being using here? You are not defining them here.

2.1.2 and 2.1.3 have the same title (PSE).

2.2 – multi class classification - - No context has been given for classification yet. What is being classified ? (presumably fault, versus no fault?).  What are the multiple classes?  What are the labels?

2.1.1 DNN- there is no need to explain the full operation and maths for a NN.  Otherwise, this paper becomes a tutorial. Also it needs to be clearer that these equations are sourced, not original.

Explain what is the implication of gradient diffusion problem – it’s not clear to the reader why this is relevant to the problem here.

2.1.2 SVMs are a well-established heavily used algorithm.  There is no need to cover all the maths and hyperplane concepts.  

2.1.3 ELM – The full title should be used to explain the acronym.  Same comments re equations – at the very least, cite sources – or else exclude.

3 Improved DST approach.  – Expand FPR acronym.  For their contribution on improved DST – is it the use of cosine similarity?  Have they referenced previous work on this (e.g. https://www.researchgate.net/publication/269073989_A_new_DS_evidcnce_fusion_algorithm_based_on_cosine_similarity_coefficient_

Of the equations 30 to 42 – which are theirs, and which are from previous sources? The text should explain clearly what is novel that is added by them, versus what is used from other sources.

3.4 – Verification . .

What “proposal” effectiveness is being tested?

Line 402 – what “modified mass function” is being referred to here? Give an equation number.

Table 1: Need to explain more clearly what equations are leading to the figures in the table.

The result of fault diagnosis in Table 2 – is “the fault model  (line 409). Explain what this means.  Explain the meaning of the results shown in table 2 more clearly.

Table 3: The title shows fusion results for different methods.  Explain more clearly the different methods. What does the Target column represent here?

The same comments apply for the comparison graphs. The meaning of the results needs to be explained.

4. Experimental Analysis  - Although Fig 6 shows the flowchart, the real purpose of the work is not given enough explanation. What exactly is the purpose of the experiment(s).   What are the classes to be tackled by the classifier?  How much data is being used to train the classifiers? Are they being used in parallel ensemble?  112 samples are mentioned for each “condition”. What is a condition here?

4.3 Classification using single stage classifier – Is this supervised learning being used here? how many classes? Also, the DNN number of hidden layers  graph – Figure 10 – why is this focussed on - when the volume of training data for weight optimisation is not explained? The methodology cannot be assessed without being clearer on the data volumes. 

The test/ training  split  - is this explained? Is it n fold cross validation?

In Graph 11(a) the meaning of B, C, IR, N etc on the Y axis needs to be explained.

Five classes are mentioned.  They need to be explained. Same for Fig 12 and 13.

Table 6 – classification accuracy – this is a straight % of the full dataset.  Is there any drill down into class level?  What is the average class accuracy?

Figure 14 – what is the unit for the Y axis fusion result?

5 -Conclusions – These can be strengthened by giving more context to the work, even in the conclusion. Also, the originality of the work over the state of the art can be restated here.  Also, is this work limited to fault diagnosis? What the future work ?

Presentation:

A full proof read and grammar check will benefit the paper,

Introduction: The paragraph, “Moreover may fuzzy modelling approaches”..  Avoid using “moreover” – it doesn’t tell the reader where you mean “as well as”, “in support of”, “despite” etc.

HCE acronym in first paragraph should be explained.

Graphs 11 (a) and (b) are too small for a print journal.

Author Response

Dear editor:

Thank you very much for giving us a chance to respond to the referees’ comments. We would like to thank the referees for their insight, which helped greatly in improving the clarity and presentation of our original manuscript. We have considered the comments of referees’ and made some changes in this revised version. Our changes and response are presented below. The point-to-point answers and explanations for all comments are listed following this letter, and the modified words and sentences are marked with red in the revised version. Thank you for your time and your consideration of this paper. I hope that the revised manuscript is now suitable for publication. If you have any question about this paper, please contact us without hesitate.

Best regards,

Yanxue Wang, Fang Liu, Aihua Zhu

Reviewer 2 Report

The authors proposed an improved Dempster-Shafer theory for the hybrid classifier ensemble approach. The results of the proposed method were improved by comparing with the ordinary methods. The detailed comments are following.

1. p.4, lines 140-156

The paragraph same as the previous page is repeated. 

2. p.7, line 170

“PSE” is “TFE” correctly

3. pp.5-8, lines 188, 235, 268

The section numbers are wrong.

4. p.13, Tables 3-4

Jiang et al. [49] suddenly appears as comparison. The brief explanation of the method is required for fair comparisons.

5. section 3.4

The authors shows only the case of “F2”. How about the other cases? It is better to describe the results of other cases in the text. The corresponding tables and figures are not necessary because of those redundancy. 

6. p.15, line 450

What is “WPT”?

7. section 4.3

The number of training data in DNN looks small. Is it enough to learn the DNN?

8. p.16, lines 469-470

It is better to explain how to set the parameters.

9. section 4.4

The reference [27] suddenly appears as comparison. The brief explanation of the method is required for fair comparisons.

10. section 5

“Future works” is missing. It should be added as research paper.

Author Response

(The authors gave the same response as above.)

Reviewer 3 Report

Using iThenticate, the similarity rate is 48%. The authors copied same texts from other publications without any changes and without citing the appropriate references.

A couple of examples: 

Line 226-230 is copied from Zhou, F., Hu, P., Yang, S., & Wen, C. (2018). A Multimodal Feature Fusion-Based Deep Learning Method for Online Fault Diagnosis of Rotating Machinery. Sensors18(10), 3521.

Line 236-238 is copied from Gao, Y., Liu, S., Li, F., & Liu, Z. (2016). Fault detection and diagnosis method for cooling dehumidifier based on LS-SVM NARX model. International Journal of Refrigeration61, 69-81.

Line 245-248 is copied from Moosavian, A., Khazaee, M., Najafi, G., Kettner, M., & Mamat, R. (2015). Spark plug fault recognition based on sensor fusion and classifier combination using Dempster–Shafer evidence theory. Applied Acoustics93, 120-129.

There are many. 

Author Response

(The authors gave the same response as above.)

Round 2

Reviewer 1 Report

The reviewer acknowledges that the review points have been somewhat addressed which has improved clarity. 

As a reviewer, there are points that still need to be addressed:

·         Avoidance of reproducing mathematical and concept theory of well documented algorithms and concepts;

·         Need to use “re-executable” methodology should include the purpose, how the work is done, S/w framework used,  evaluation metrics, datasets  - all the components needed for other researchers to re-execute the work if they needed to.

·         The experimental results – are mixed in with methodology – e .g. training approach, labels etc.

·         References need to be checked as per comments.

·         Overall proof reading of English

Where “Ref” is used below, it is referring to the point # used by the reviewers in their first response.

________________________________________________________________________________

Ref Point 1:  The sentence added to provide context:   Rolling element bearings are the key components widely used in rotating machines. However, 29 they are easy to be damaged because of their harsh running conditions. An unexpected failure of the 30 rolling element bearings may cause the sudden breakdown of the system, even a severe catastrophe

This sentence still needs more qualification to explain the context. As a journal paper, the authors should not be reluctant to reference the practical real world domain and application of rotating machines.   What are they, why are they used? What are the harsh conditions referenced?  Some rxtra sentences will give readers who are interested in your work but not familiar with your domain, a better chance to understand your work and cite your paper.

New sentence

Nevertheless, the accuracy of the existing techniques still needs to be improved using the existing techniques based on the patterns generated from a single information source due to its 49 complicated structure.

This sentence is not meaningful without commas to explain it.  It needs to reworded .

Point 3: It can also be found that DST plays a crucial role in decision fusion.

This sentence hasn’t been reworded as per previous feedback  (although conflicting evidence point follows it)

Ref Point 4: The “lack of robustness” comment on DST – is this referring to the problem of highly conflicting evidence sources?  Or is it something else?

Text change is required to remove this ambiguity – rather than just answering for the reviewer.

Ref Point 6 – References are needed for the new text mentioning “Jousselme distance  and MaxDiff distance”

Ref Point 7: The final sentence “These methods can reduce the influence of unreliable evidence sources”.

The text change does not go far enough to explain the point raised.

Ref Point 8 – the example in Table 3 is useful for showing the proposed method improves over murhy, zhang etc.  Why is it redundant? If this improvement is being claimed as a contribution in this paper, it needs to be supported with evaluation data in this paper.

Point 9  - doesn’t go far enough to address the original feedback. Why these features for HCE? More information on the purpose and context of the work needed. 

The methodology section should in theory allow another follow on researcher to repeat the work, knowing what to do, the evaluation metrics, the experimental work approach, the order of experiments etc.   

Also the sentence here is ambiguous and lacking in detail:

“Due to the nonlinear and nonstationary of rolling bearing signals, some feature extraction techniques have been thus developed to improve the accuracy of  fault diagnosis, such as various information entropy methods given in [47].

  What are “some feature extraction techniques”.. a variety?  By this research or others? Why does the non linearity and movement of the signals require feature extract techniques?

Ref Point 12:  The authors have not added the source to the equations – they have numbered the equations, which is different. The sources are now included in the text – but it needs to be made clearer that the equations are coming from the original source e. g. As explained in {xx], the calculation for YY is shown as..  

Also – can the authors explain why they are reproducing the equations here? What value are they adding ?

Paragraph 2.2 “In this Section, three classical multi-classifier models are introduced, including Deep…” .  

Clarify  - is this sentence saying that deep learning, svm and ELM are multi-classifier model?  Do they mean multi-class?  SVM for example is a stand along classifier. There is ambiguity or mis use of terms here.

Section 2.2.1 – Deep learning NNs:

It is not necessary to explain deep NNs – and the underlying maths – this is done many times before.

The reference for the maths associated with the DNN is reference 48 – which is the application of a DNN.  Is this a correct reference? It seems very specific as a source for these equations.  See comments re SVM equations in next point – also applies here.

2.2.2 SVMS section – the reference 49 does not appear to be correct as a source for SVM optimisation and kernel function.   As before – unless these equations are being manipulated or directly used    - there is no justification for reproducing them here.

Section 3:.4 – Verification with numerical analysis –  Is it the case that the approach is verified with a worked example, as opposed to experimental work?

“The practical data in [45] was adapted for the convenience of making a comparative study”.. how was it adapted? Comparative with [45]? 

“BPAs from [44] were directly adopted.. more details can be found in [44] “  It would be worth explaining these details  (as opposed to have used up so many pages with unchanged equations from other works ?).  The reader needs the help to understand these.

Suppose there were three types of fault in a motor rotor, denoted as ?1={????? ?????????}, 334 ?2={????? ????????????}and?3={???????? ?????????}, respectively. Three vibration accelerometer sensors were installed in different positions to collect the vibration signals”

In the sentence above, why “suppose”? Is this a made up example?   Were vibration accelerometers set up or not?  What is real experimental work or not? 

4. Experimental analysis – “Its effectiveness and robustness would be further demonstrated 369 using our practical roller bearing experimental datasets”.. has this been done?  If so -  change “would” to “is”  - it completely changes the meaning of the sentence.

4.1 experimental set up – MFS-MG test rig is declared as it is has been already explain. Need details. what is MFS-MG etc.

Figure 10 – is classification accuracy across the whole dataset – as hinted in the text as “total classification accuracy”?  Or average accuracy across the 5 classes? It should be the latter.  And whichever it is should be explained.  Also, the methodology section should have explained which evaluation metric was being used.

Lines 436 onwards – listing out various class accuracies – better to either put these in a diagram – or use average class accuracy – or use a confusion metric.  Are the five “conditions”  the same as the five classes (labels?).

What is the class balance in the dataset of the 5 classes?

Training versus test data:  “The training data sets include 420 rows and 4 columns, the testing data sets included 140 rows and 4 columns.” Why this split? How were they selected?  Is the training data labelled? And if so, it has an extra column?

At no point is the term supervised learning used – but this is the technique used in the paper.  It needs to be included.

Table 6: presentation of results – should be using average class accuracy – not overall dataset accuracy. Otherwise, individual classes may be performing extra poorly or well, and this is hidden.

Fig 14 15 16 – what does “number of evidence” mean in the x-axis? .

Author Response

(The authors gave the same response as above.)

Reviewer 3 Report

Plagiarism is detected. Line 212-215 is completely copied from another paper. 

Author Response

(The authors gave the same response as above.)

Round 3

Reviewer 1 Report

Approved  - subject to proof reading to remove grammar errors and readability. 

Author Response

Reviewer: Approved  - subject to proof reading to remove grammar errors and readability. 

Authors: Thanks for your comments. We have thoroughly checked our work again and remove several grammar errors.